# Temperature, Pressure, and Velocity Influence on the Tribological Properties of PA66 and PA46 Polyamides

**DOI:** 10.3390/ma12203452

**Published:** 2019-10-22

**Authors:** Mihai Tiberiu Lates, Radu Velicu, Cornel Catalin Gavrila

**Affiliations:** Mechatronics and Environment Department, Product Design Faculty, Transilvania University of Brasov, 500036 Brasov, Romania; rvelicu@unitbv.ro (R.V.); cgavrila@unitbv.ro (C.C.G.)

**Keywords:** wear, friction, tests, polymers, coefficient of friction

## Abstract

The tribological properties of PA66, PA46, and PTFE-mixed PA46 were investigated. The tests were achieved on a pin-on-disc tribometer. Before tests with different sets of parameters, a running-in-type test (with constant parameters) was performed for all the materials, under lubricated and dry conditions. The tests parameters were variable referring on load, velocity, and temperature. The results are referring on the value of the wear developed during the run-in period and on the variation of friction coefficient with the testing parameters. The results show that the PTFE-mixed PA46 polyamide has better tribological properties than the PA66 and the PA46 polyamide.

## 1. Introduction

The tribological properties influence the durability and reliability of the machine elements which are used in sliding or rolling friction conditions. The development of the polyamides in the last years is offering new materials with high tribological properties such as wear resistance and small friction coefficients. These polyamides, such as PA66, PA46, and modified PA46, are used in construction of mechanical parts such as gears, cams, guides, and ball screws which work in different conditions of temperature, lubrication, pressure, and velocity.

In the literature, the tribological properties of the polyamides are studied by tests performed on reciprocating or pin-on-disc tribometer modules; as a result, the wear and the friction coefficients in different testing conditions are determined.

An application of the polyamides in gears is presented in [1] were the wear in rolling–sliding contact is studied and it is certified that the studied material has good wear resistance; good wear resistance is obtained also for the PA blend type polyamides studied in [2]. The polyamides’ tribological properties and their applications are studied in [3]; according to this, polyamide 6 and also graphite and wax polyamide 6 have applications in aerospace, automotive, electronics, and chemical industries because these materials have high strength/weight ratio, self-lubricant properties, good damping properties, corrosion resistance, and simple and economic fabrication process. Good tensile properties of PA6 and Al_2_O_3_-reinforced PA6 polymer composites are determined in [4]. The wear properties of PA-type polymers are studied in [5,6] and their wear resistance is highlighted.

The influence of the temperature on the tribological properties of glass fiber MoS_2_-reinforced PA66 journal bearings is studied in [7]. The paper presents the influence of the sliding velocities, temperature, and pressure on wear of journal bearings made of PA66 reinforced with glass fiber and MoS_2_. The increase of the temperatures, pressure, and velocities increases the wear. The tests were performed for a journal bearing with sliding velocities of 0.5 and 1 m/s for 1 km sliding distance; the pressure was at 0.095 and 0.190 MPa. A problem of the PA66 polyamide is identified also in [8] as a high sensitivity of its tribological properties with the increasing of the temperature.

The tests performed on the reciprocating tribometer presents as results, the variation of the friction coefficient and of the wear depending on the sliding velocities and the normal load [9,10].

The paper [9] presents the study of the friction coefficient and of the wear in the case of a GCr15 steel ball in contact with a PA66 plate; the normal loads were 1, 2, 3, and 4 N for a diameter of 4 mm for the ball and the sliding velocities were 31.42, 62.83, 94.25, and 1245.66 mm/s. The conclusion of the paper is that under dry conditions, the friction coefficient decreases with the increase of the normal load with values between 0.4 and 0.15 and increases with the increase of the translational speed. The friction coefficient has a stable evolution after 400 s of tests.

In [10], the tribological behavior of PA66 under dry sliding conditions is studied. Tests were performed on a UMT reciprocating tribometer, ball on block, at a humidity of 40% with a stroke of 5 mm, for five tests, by taking the average of the results. The test period is about 1800 s and the friction coefficient of PA66 neat is in the interval of 0.22–0.28 increasing during test time. For a stroke frequency of 4 Hz and loads of 6, 9, 12, and 15 N, the friction coefficient increases with the increase of the load. For a load of 9 N and strokes frequencies of 1, 4, 8, and 12 Hz, the friction coefficient increases with the increase of the frequency.

According to the literature, the usual tests for the study of the tribological properties of the polyamides have been performed on pin-on-disc tribometers.

The paper [11] presents the mechanical properties of a class of PA66 polyamides. The tests are performed on a pin-on-disc type tribometer under dry and lubricated conditions. In the case of no lubricated conditions, the friction coefficient increases with the increase of the force and has values between 0.15 and 0.23 for forces between 50 and 250 N, the pin diameter of 10 mm, and velocities of 0.025 m/s. The friction coefficient decreases with the increase of the sliding velocity—below 0.1 for 0.1 m/s. In the case of lubricated conditions, the friction coefficient decreases with the increase of the normal force (with values between 0.05 and 0.06) for forces between 50 and 250 N.

In [12], the tribological behavior of polyamide 66 and of polyamide 66 composite filled with rice bran ceramic particles at a wide range of pressure–velocity values under dry conditions is presented. The specific wear rates of pure PA66 changed with the normalized surface temperature because the wear mode transited. The tests have been performed on a pin-on-disc type tester and the friction coefficient was calculated from a torque measured ring using a torque transducer coupled to the stage. The input data of the tests are represented by increasing values of contact pressures and sliding velocities; PA66 shows increasing values of the friction coefficient between 0.2 and 0.65 when the product between the normal pressure and the velocity Pv exceeds 2.5 MPa m/s. PA66 has high wear rates at low Pv values; these values decrease with the increase of Pv values.

In [13], the wear is studied using a pin-on-disc module with normal loads of 5, 10, 20, and 30 N and rotational speeds of 1000, 1500, and 2000 rpm for a pin of 12 mm diameter made of PA and a steel disk in dry friction conditions. The friction coefficient decreases with the increase of the force with values between 0.14 and 0.42 with 0.32 and 0.42 for the pure PA66 and the other values for the graphite-reinforced PA. At the beginning, the wear is high due to the roughness of the surfaces being in contact; after that, during tests the surfaces are smoothed; higher wear rates are obtained for the pure PA66.

The tribological properties of the PA46 polyamide have been investigated in [14]; according to the results, the PA46 polyamides have a 20%–30% lower sliding friction coefficient than the PA66 polyamide in tests performed on a rotary module with speeds between 1000 and 4000 rpm, normal pressures between 0.5 and 5 MPa, and temperatures between 125–160 °C.

The tests performed in [15,16] show that the PA46 polyamides conserve their frictional properties better than the PA66 polyamides at high temperatures (210 °C). The results from [17,18] conclude that the PA46 polyamides have smaller wear than the PA66 polyamides, for all the test conditions.

The paper [19] studies the fractal behavior of sliding surfaces of PA66 polyamide with glass fiber reinforcement by using tribological tests. The multi-fractal method used in the paper allows the understanding of the relationship between the morphological structure, the tribological parameters, and the surface roughness. The same authors from paper [19] develop a methodology based on digital image processing techniques, in paper [20], in order to analyze the morphological aspects of the sliding surfaces for two types of polyamides: the fiber-glass-reinforced PA66 and the MoS_2_-filled PA66. The results show that the increase of the velocity increases the frictional properties of the fiber-glass-reinforced PA66.

The aim of the paper is to evaluate the tribological properties (the wear and the friction coefficient) of PA66, PA46, and PTFE-mixed PA46 by performing tests on a pin-on-disc tribometer, with different temperatures, pressures, and velocities. The obtained results are useful for the researchers and for the industries which work with polyamide parts being in contacts with steel parts in relative motion such as gears, cams, guides, and ball screws which work in different conditions of temperature, lubrication, pressure, and velocity. These results indicate which types of polyamides may be used in certain exploitation conditions.

## 2. Materials and Methods

The tests were performed for three types of polyamides plates. One of the materials is the PA66 polyamide obtained by injection molding, unreinforced, and heat stabilized with a humidity absorption of 2.5%, tensile modulus of 3000 MPa, deflection temperature at 1.8 MPa, stress of 75 °C, and melting temperature of 260 °C [21]. The second material is the PA46 polyamide obtained also by injection molding, unreinforced, and heat stabilized with a humidity absorption of 3.4%, tensile modulus of 2800 MPa, deflection temperature at 1.8 MPa, stress of 190 °C, and melting temperature of 295 °C [21]. The third material is a PTFE-mixed PA46 polyamide, heat stabilized and friction modified with a humidity absorption of 3.2% and a tensile modulus of 3100 MPa [21].

The tribometer used for the tests is a pin-on-disc type as presented in Figure 1; the force transducer of the tribometer allows measurement of the normal load, *F_n_*, and the tangential load, *F_t_*, up to 1000 N with a resolution of 50 mN [22]. The steel 416 stainless steel pin has a diameter of 6.3 mm, a length of 25.4 mm, a hardness of RC 38, is mounted in a holder, and is acting on a disk where the rotational speed, *n*, can be adjusted between 0.001 and 5000 rpm in two directions [22].

The heater can heat the oil bath up to 150 °C [22]. The wear is measured directly by the vertical positioning sensor and the friction coefficient is determined as the ratio between the tangential load, *F_t_*, and the normal load, *F_n_*.

The polyamide plates used for tests are fixed by using a bolt in a milled slot in the rotating disk, as can be observed in Figure 2. These sample plates have been cut from polyamide sheets. The sliding radius is equal to 15 mm.

The tests follow two steps; first a running-in-type test is performed with constant test parameters in lubricated and in dry conditions for a period of 1 h, with a normal load of 50 N (equivalent to a normal pressure of 1.76 MPa), a rotational speed equivalent to a velocity of 1.25 m/s, at a room temperature of 20 °C, and an environmental humidity of 42%.

The second type of tests are achieved for normal loads of 4.5, 45, 70, and 90 N which are equivalent to normal pressures of 0.159, 1.591, 2.475, and 3.183 MPa. The rotational speeds are equivalent to linear velocities of 0.025, 0.2, 0.5, 1, and 2 m/s. The temperatures are represented by the room temperature of 20 °C and the oil bath temperature of 60 and 90 °C. All these test parameters are presented in Table 1. The tests are made in lubricated conditions at an environmental humidity of 42%.

## 3. Results

For the lubricated and dry conditions, the wear developed during the running-in with constant test parameters was measured—Figure 3. The wear was calculated by subtracting the value of the vertical position of the pin with normal load, before the running-in process, from the value of the vertical position of the pin, also under normal load, at the end of the running-in process. This includes differences in deformation of disc and pin, due to change of temperature. According to the results, the PA66 polyamide has the smallest wear in lubricated conditions but it fails very fast in dry friction conditions; the PA46 and the PTFE-mixed PA46 have almost the same wear in dry friction conditions.

A short discussion is needed here. The lubricated case of running-in and also of the rest of the tests only creates conditions for boundary lubrication since the geometry of the contact is not proper for hydrodynamic lubrication. The results about wear, presented in Figure 3 should be analyzed according to Archard theory, representative for the wear trend, which stipulates that the wear is inversely proportional to the hardness of the material being worn away [23]. PA46 has similar mechanical characteristics to PA66 at room temperature. All three materials are subject to mechanical softening at high temperature. An important fact is that PA66 has a lower melting temperature and faster mechanical softening with increasing temperature, given also by the smaller deflection temperature at 1.8 MPa stress.

Another very important aspect for this discussion is the temperature of the contact surface. As shown in [24], for a dry pin-on-disc test, steel pin on a soft metal Sn disc, the temperature at the contact surface of the pin reaches 300 °C. Without measuring this temperature, we assume that, in our case, high temperatures may be obtained at the contact surface of the pin. As a result, the higher wear of PA66 in dry condition may be explained by the very high temperature of the pin creating important melting wear of PA66 in comparison with PA46. According to [25], very fast wear in a polymer due to exceeding a critical temperature has consistent experimental evidence. In order to explain the different results in the case of lubricated condition, the influence of the lubricant with different absorption percentage, also acting as cooling environment, and avoiding melting wear should explain the smaller wear of PA66. The difference between PA46 and PTFE-mixed PA46 may be explained with different wear mechanism involving the different grain structure of PTFE-mixed PA46.

For the lubricated conditions, Figure 4, Figure 5 and Figure 6 present the variation of the friction coefficient for the PA66, PA46, and PTFE-mixed PA46 polyamide with the temperature, for different normal pressures and speeds; the friction coefficient increases with the increase of the temperature (a highest increasing is in the case of small normal pressure), for the tested materials. Small values of the friction coefficient are obtained for the PTFE-mixed PA46 polyamide and it is more stable in the case of the normal pressure variation.

Figure 7, Figure 8 and Figure 9 present the variation of the friction coefficient for the PA66, PA46, and PTFE-mixed PA46 polyamide with the normal pressure, for temperatures equal to 20, 60, and 90 °C, in lubricated conditions, at different speeds; the friction coefficient decreases with the increase of the normal pressure. The smallest values of the friction coefficient are obtained in the case of PTFE-mixed PA46 polyamide; at high normal pressures, the pressure coefficients for the tested materials are very close in value. According to the variation of the friction coefficient with the normal pressure, the PTFE-mixed PA46 polyamide has the smallest variation, so is more stable than the other two materials. For the PA66 and PA46 polyamides there is a decreasing tendency of the friction coefficient with the increase of the normal pressure up to 2.5 MPa; for bigger values of the normal pressure the friction coefficient has a small increase.

The variation of the friction coefficient with the velocity, for different normal pressures, in the case of PA66, PA46, and PTFE-mixed PA46, in lubricated conditions, is presented in Figure 10, Figure 11 and Figure 12 for temperatures of 20, 60, and 90 °C; the friction coefficient decreases with the increase of the velocity. For all the cases, the PTFE-mixed PA46 polyamide has the smallest values of the friction coefficient; bigger differences between the friction coefficients of different polyamides are obtained in the case of high temperatures—90 °C.

The evolution of the friction coefficient with speed, temperature, and load should be explained based on Stribeck friction theory [23].

For boundary lubrication, according to Stribeck law, increasing speed determines decreasing friction coefficient as it is generally shown in Figure 10, Figure 11 and Figure 12.

Increasing oil temperature determines decreasing oil viscosity and according to Stribeck law for boundary lubrication leads to increasing friction coefficient. This also validates the results presented in Figure 4, Figure 5 and Figure 6.

A different situation is in the case of load (pressure) influence on friction coefficient. According to Stribeck theory [23] for boundary lubrication, increasing load should lead to increasing friction coefficient. The results presented in Figure 7, Figure 8 and Figure 9 are opposite to Stribeck friction theory, but confirmed in [11], for PA66. This can be explained by modified properties of the polyamides. Increasing load creates worse lubrication conditions when looking at the oil film. In the same time, the temperature of the pin surface in contact with the PA disk increases leading to mechanical softening at the superficial layer of the polyamides. Reduction of shearing strength will lead to decreased friction coefficient. The influence is bigger between normal pressures of 0.159 and 1.591 MPa, when the load increases 10 times, in comparison with the next stage, from 1.591 to 3.183 MPa, when the load only increases two times.

## 4. Conclusions

The friction coefficient of the PA66, PA46, and the PTFE-mixed PA46 polyamides decreases with the increase of the normal pressure and of the velocity; the value of the friction coefficient increases with the increase of the temperature. The PTFE-mixed PA46 polyamide has the smallest friction coefficient in all the testing conditions, and it has a smaller sensitivity with the variation of the normal pressure, velocity, and temperature.

The PA66 and the PA46 polyamides have a decrease of the friction coefficient with the increase of the normal pressure up to 2.5 MPa; for bigger values of the normal pressure, the friction coefficient has a small increase.

In lubricated conditions, the PA66 polyamide developed the smallest wear during the run-in period; the PTFE mixed polyamide developed the highest wear. In dry friction condition, at the same loading and speed, the PA66 polyamide was destroyed and the PA46 and PTFE-mixed PA46 polyamide developed almost the same wear.

## Figures and Tables

**Figure 1 materials-12-03452-f001:**
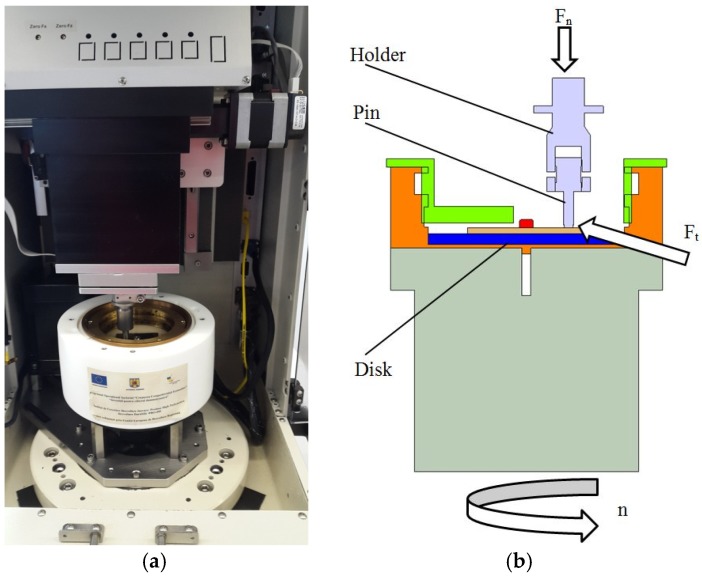
The tribometer. (**a**) Front view; (**b**) cross section.

**Figure 2 materials-12-03452-f002:**
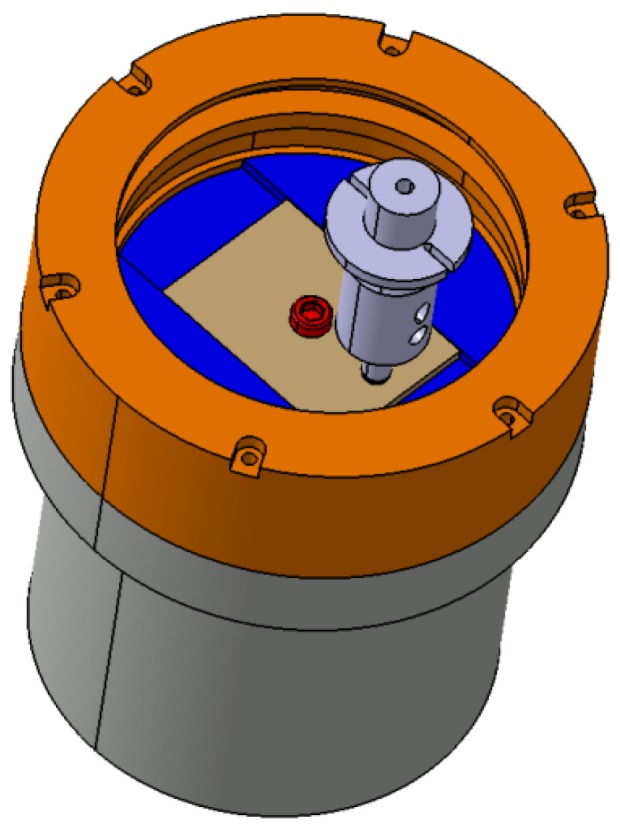
The polyamide plate mounted in the rotating disk.

**Figure 3 materials-12-03452-f003:**
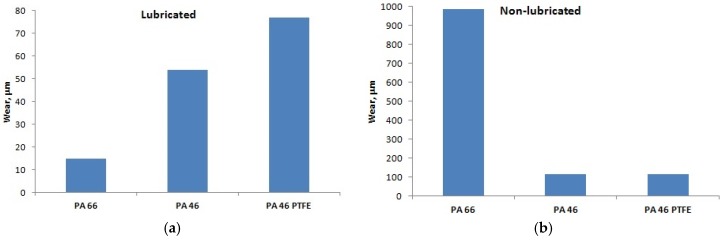
The wear during running-in. (**a**) Lubricated; (**b**) non-lubricated.

**Figure 4 materials-12-03452-f004:**
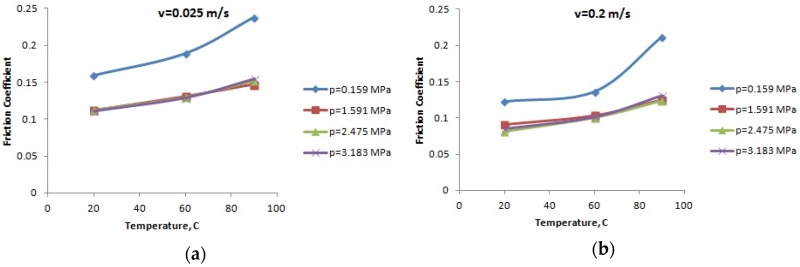
The variation of the friction coefficient of PA66 with the temperature: (**a**) speed *v* = 0.025 m/s; (**b**) speed *v* = 0.2 m/s; (**c**) speed *v* = 0.5 m/s; (**d**) speed *v* = 1 m/s; and (**e**) speed *v* = 2 m/s.

**Figure 5 materials-12-03452-f005:**
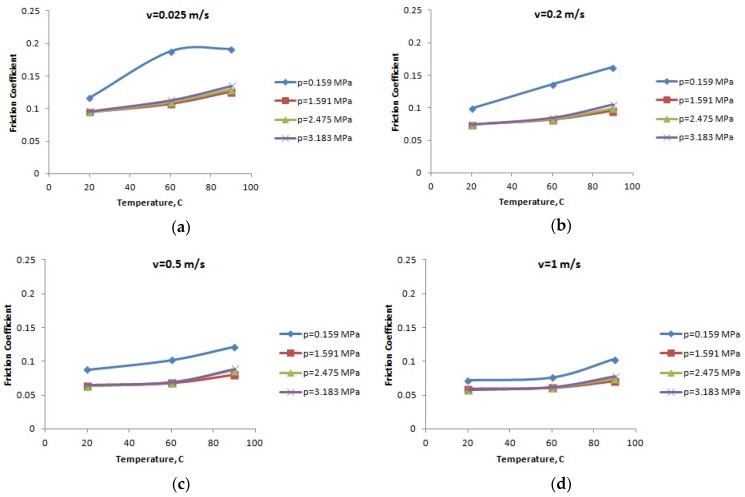
The variation of the friction coefficient of PA46 with the temperature: (**a**) speed *v* = 0.025 m/s; (**b**) speed *v* = 0.2 m/s; (**c**) speed *v* = 0.5 m/s; (**d**) speed *v* = 1 m/s; and (**e**) speed *v* = 2 m/s.

**Figure 6 materials-12-03452-f006:**
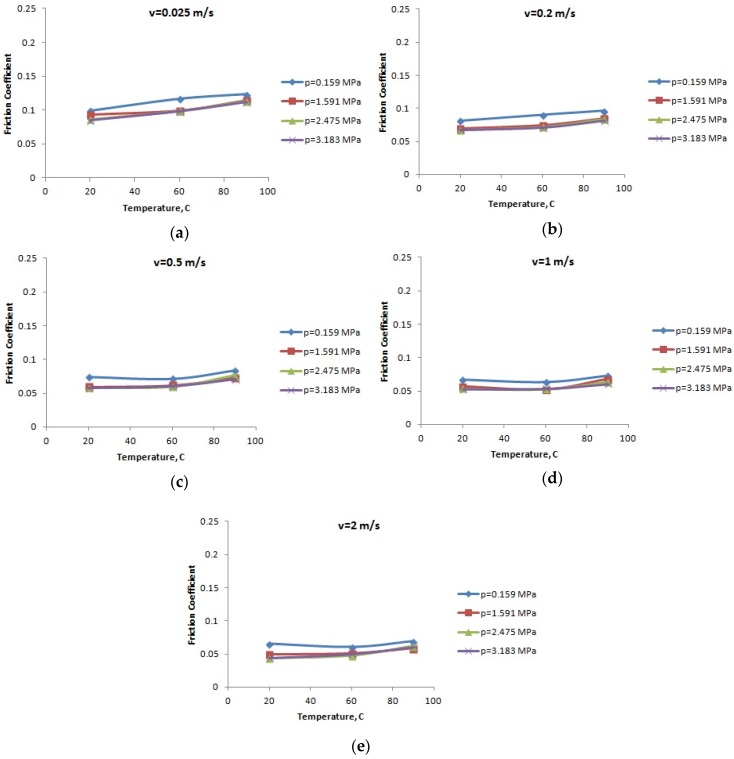
The variation of the friction coefficient of PTFE mixed PA46 with the temperature: (**a**) speed *v* = 0.025 m/s; (**b**) speed *v* = 0.2 m/s; (**c**) speed *v* = 0.5 m/s; (**d**) speed *v* = 1 m/s; and (**e**) speed *v* = 2 m/s.

**Figure 7 materials-12-03452-f007:**
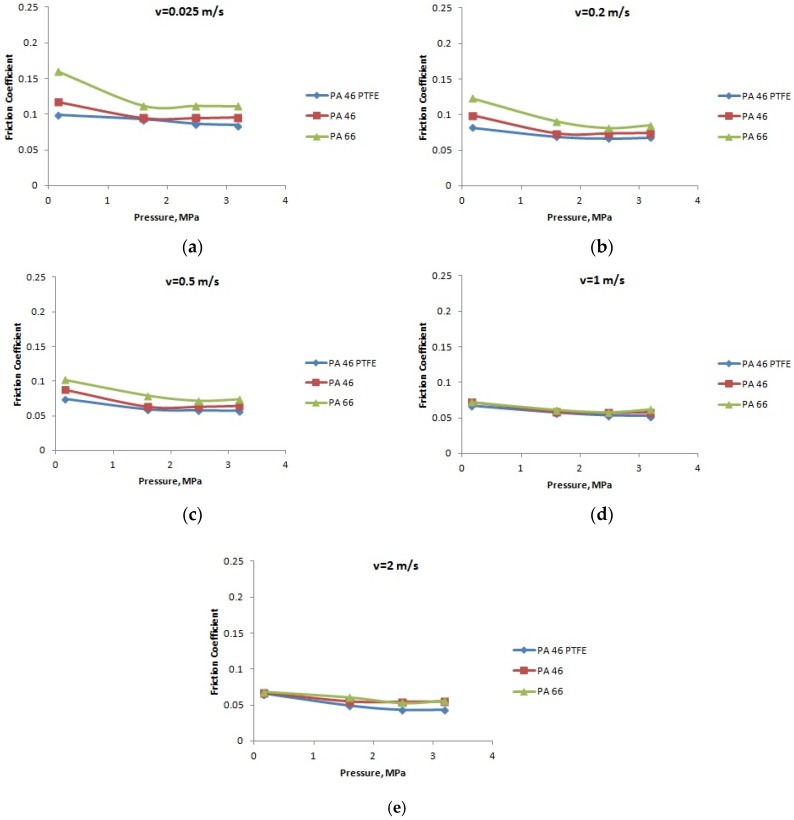
The variation of the friction coefficient with the normal pressure, at a temperature of 20 °C: (**a**) speed *v* = 0.025 m/s; (**b**) speed *v* = 0.2 m/s; (**c**) speed *v* = 0.5 m/s; (**d**) speed *v* = 1 m/s; and (**e**) speed *v* = 2 m/s.

**Figure 8 materials-12-03452-f008:**
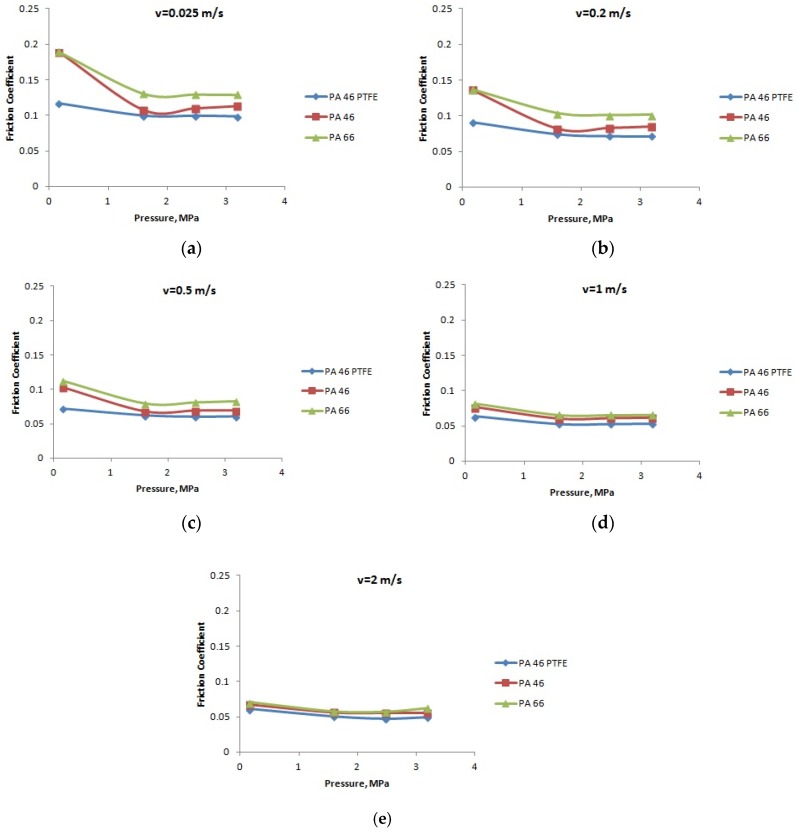
The variation of the friction coefficient with the normal pressure, at a temperature of 60 °C: (**a**) speed *v* = 0.025 m/s; (**b**) speed *v* = 0.2 m/s; (**c**) speed *v* = 0.5 m/s; (**d**) speed *v* = 1 m/s; and (**e**) speed *v* = 2 m/s.

**Figure 9 materials-12-03452-f009:**
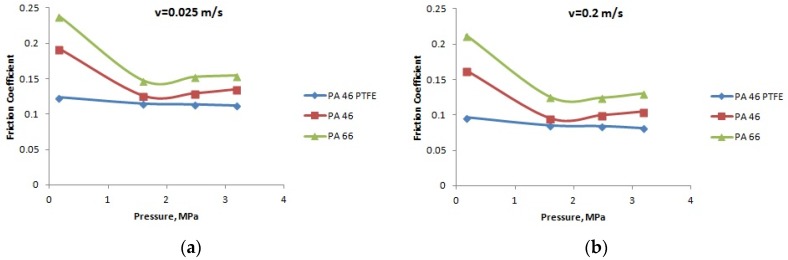
The variation of the friction coefficient with the normal pressure, at a temperature of 90 °C: (**a**) speed *v* = 0.025 m/s; (**b**) speed *v* = 0.2 m/s; (**c**) speed *v* = 0.5 m/s; (**d**) speed *v* = 1 m/s; and (**e**) speed *v* = 2 m/s.

**Figure 10 materials-12-03452-f010:**
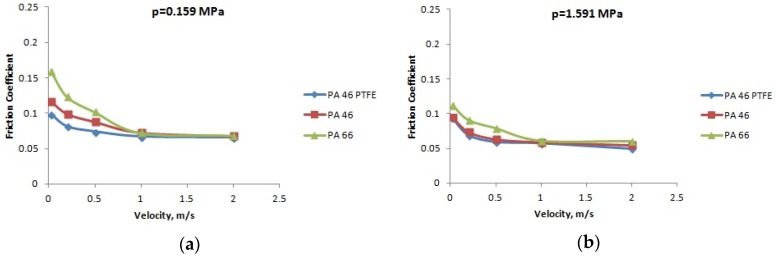
The variation of the friction coefficient with the velocity, at a temperature of 20 °C: (**a**) pressure *p* = 0.159 MPa; (**b**) pressure *p* = 1.591 MPa; (**c**) pressure *p* = 2.475 MPa; and (**d**) pressure *p* = 3.183 MPa.

**Figure 11 materials-12-03452-f011:**
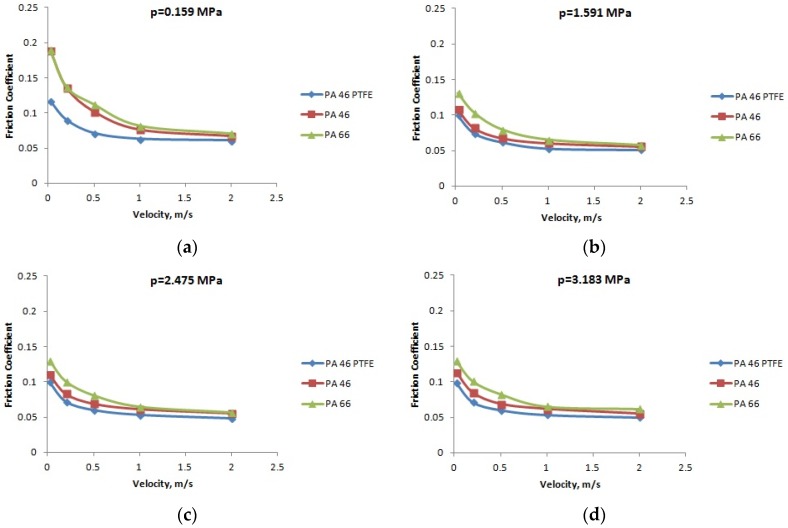
The variation of the friction coefficient with the velocity, at a temperature of 60 °C: (**a**) pressure *p* = 0.159 MPa; (**b**) pressure *p* = 1.591 MPa; (**c**) pressure *p* = 2.475 MPa; and (**d**) pressure *p* = 3.183 MPa.

**Figure 12 materials-12-03452-f012:**
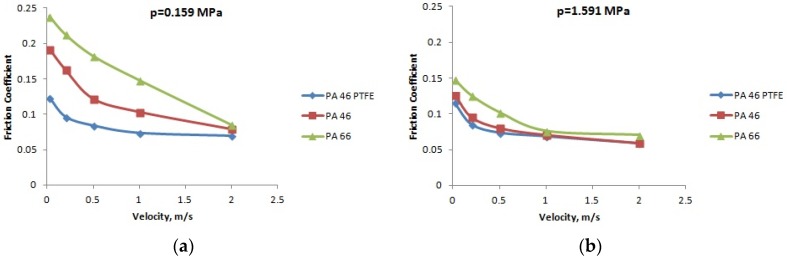
The variation of the friction coefficient with the velocity, at a temperature of 90 °C: (**a**) pressure *p* = 0.159 MPa; (**b**) pressure *p* = 1.591 MPa; (**c**) pressure *p* = 2.475 MPa; and (**d**) pressure *p* = 3.183 MPa.

**Table 1 materials-12-03452-t001:** The test parameters.

Polyamide	Pressure (MPa)	Speed (m/s)	Temperature (°C)
PA66	0.159	0.025	20
PA46	1.591	0.2	60
PA46 PTFE	2.475	0.5	90
	3.183	1	
		2

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
