# Peer review of "Temperature, Pressure, and Velocity Influence on the Tribological Properties of PA66 and PA46 Polyamides"

_materials, 2019, doi:10.3390/ma12203452_

Round 1
Reviewer 1 Report
In the abstract and many more places Teflon has to be replaced with PTFE, Teflon is a brand name from DuPont. - pin-on-disk is usually "pin-on-disc" and the abbreviation is PoD in tribology literature in many cases. In the intro not necessary to write about PA6, the examined materials are PA66 and PA46 From the intro the role of humidity on the tribo-mechanical properties is missing and even later, the preparations of PA samples does not contain the conditioning or drying…etc. - About the test layout: the sliding radius of the path is not given. As I mentioned, the preparation of the sample is not reproducible. - To understand the behaviour of the measured curves the exact material properties (mechanical, and surface energy) are essential. I can not find it. The composition of PA46/PTFE composite is unknown. - Also problematic with this type of tribotest systems the “wear”. What the article call as wear, that is really wear+deformation together. They have to be separated, or, must use the sign that “wear+deformation” . - Figure 2. has no information concerning the target of the article. In Figure 3. The coloumns refer of “wear” values for the given three type of polymers. How was one exact value extracted from the online measured wear curve? The same question rose up for the identical friction coefficient values later e.g.in Fig 4. - The scientific discussion is missing about the Fig. 4-9. The description of the measurements as “decreasing and increasing…..” are measured facts but the phenomena behind the data are missing. The minimum expectation is the evaluation of the tribological results from the point of Archard theory analysing the deformation and adhesion effects of friction. That is why the mechanical and surface energy values are necessary data. In the light of those the conclusion will rather different.
Author Response
Please see the attachment called Review1

Reviewer 2 Report
In the manuscript “Temperature, pressure and speed influence the Tribal Properties of PA66 and PA46 polyamides ”the tribological properties (friction and wear) of PA 66, PA 46 and Teflon Mixed PA 46 were investigated. The parameters: load, speed and temperature were related to wear and friction in tribological pairs. The results shown in teflon-blended PA 46 polyamide have the best tribological properties of PA 66 and PA 46 polyamide. The study of polyamides and their compounds is of great interest from an industrial point of view. The work has a large number of tests apparently done properly. However, several aspects must be improved in this manuscript before they can be accepted for publication: In the fourth paragraph of the authors' introduction, a work that studied the influence of sliding velocity on PA, PA reinforced with glass fibers and PA with the addition of MoS2. There are a number of recent articles that have also studied these relationships, including * A table containing the test conditions should be included in the methodology topic for ease of understanding. The last paragraph of the introduction should be rewritten. You must contain the purpose of the work and the overall results.
*3-D reconstruction by extended depth-of-field in tribological analysis: Fractal approach of sliding surface in Polyamide66 with glass fiber reinforcement, Horovistiz, A., Laranjeira, S., Davim, J.P., Polymer TestingVolume 73, February 2019, Pages 178-185.
*Influence of sliding velocity on the tribological behavior of PA66GF30 and PA66 + MoS2: an analysis of morphology of sliding surface by digital image processing, Horovistiz, A., Laranjeira, S., Davim, J.P., Polymer BulletinVolume 75, Issue 11, 1 November 2018, Pages 5113-5131.
Author Response
Please see the attachment called Review2

Round 2
Reviewer 1 Report
in section 2. Materials and methods. The filling ratio of PA46 composite is not given yet. What was the PTFE content? also at this section: with this testing layout the "wear" is not the real wear. As the vertical displacement is considered to be wear, that is not true. That is the sum of the wear and the deformation of the polymer disc together. in line 156 still using "disk" were the contact or contact-close temperutre measured? In case not, any prove with that about the results is just conditional (guess). point 4. should be conclusion, discussion goes with the interpretation of the results.
Reviewer 2 Report
The questions were answered / corrected properly.
Author Response
Answers to Reviewer nr.2 – Round 2
Thank you.
Kind regards.